# How competition can drive allochronic divergence: A case study in the Marine Midge, *Clunio marinus*

Alexander G. G. Jacobsen[1]*, Tobias S. Kaiser[1]*, Chaitanya S. Gokhale[2,3]*

1 Max Planck Research Group "Biological Clocks", Max Planck Institute for Evolutionary Biology, Plön, Germany, 2 Research Group for Theoretical Models of Eco-evolutionary Dynamics, Department of Theoretical Biology, Max Planck Institute for Evolutionary Biology, Plön, Germany, 3 Center for Computational and Theoretical Biology, Julius-Maximilians University, Würzburg, Germany,

☙ Joint last authors
* jacobsen@evolbio.mpg.de (AGGJ), kaiser@evolbio.mpg.de (TSK),
chaitanya.gokhale@uni-wuerzburg.de (CSG)

## Abstract

Synchronizing mating to extrinsic environmental cycles can increase the chance of successful reproduction, and the resulting temporally-assorted mating may precipitate speciation if coupled with divergent selection. This process might be particularly relevant to the marine midge *Clunio marinus*, which synchronizes its reproduction to different lunar phases. In Roscoff (France) two sympatric populations differ in reproductive timing but are still connected by gene flow. A previous study found a relationship between the timing of reproduction and the depth at which larva live in the intertidal zone. Building on this observation, we ask if the link between reproductive timing and depth could be a mechanism for divergence when coupled with competition-induced density-dependent fitness. We devise an individual-based model replicating the reproductive behavior of *C. marinus* and find that sympatric divergence can occur, even when we model sexual reproduction with recombination and an explicit genetic basis. Our results suggest this mechanism is a likely hypothesis for the allochronic divergence observed in the Roscoff populations. Additionally, our study provides insights into how density-dependent fitness and competition may play a role in allochronic divergence in general.

## Author summary

Reproduction timed to extrinsic environmental cycles is common across the tree of life, and can even result in speciation if reproductive timing were to diverge within a population. Though this has occurred for a handful of organisms, an understanding of the mechanism that might drive divergence in reproductive timing is missing. Here, we investigate this phenomenon in a marine midge, *Clunio marinus*, which has split into two

**Data availability statement:** All of the code for running the simulations and generating the figures can be found here: https://github.com/AlecJacobsen/Clunio_speciation.

**Funding:** The study was supported by a PhD fellowship of the IMPRS for Evolutionary Biology awarded to AGGJ and an ERC Starting Grant (Nr. 802923) awarded to TSK. The funders had no role in study design, data collection and analysis, decision to publish, or preparation of the manuscript.

**Competing interests:** The authors have declared that no competing interests exist.

timing types reproducing at different points in the lunar cycle. By simulating *C. marinus'* reproductive ecology, we find that divergence is a common by-product of their mating behavior, so long as the larvae are competing for space. Our results are robust to the explicit inclusion of sexual reproduction and a genetic basis for reproductive timing. This suggests that larval competition in conjunction with *C. marinus'* mating behavior is a likely mechanism driving the divergence in timing and hints at how competition might play a role in speciation via reproductive timing more generally.

## Introduction

The idea that species can diverge in the face of gene flow has been gaining traction over the last few decades as advances in sequencing and phylogenomics uncover increasingly more examples where a strictly allopatric model does not seem to be the case (e.g., [1–3], among others). As such, it has become important to understand the selective and genetic mechanisms that can facilitate divergence with gene flow, and in this regard, examples of incipient divergence, where these selective and genetic forces still act upon populations, are especially valuable. One such case is in the intertidal midge, *Clunio marinus*, which is notable for timing its development and reproduction to the lunar cycle [4]. Along the coast of Brittany, France, populations of *C. marinus* appear to have recently diverged with gene flow, resulting in several sympatric chrono-types, or subpopulations that differ in their reproductive timing [5]. These differences in timing restrict gene flow between chronotypes [5], suggesting that *C. marinus'* repro-ductive timing acted as a multiple-effect or magic trait [6]. However, what selective and genetic factors drove or facilitated this divergence remain a mystery.

Divergence in reproductive timing (termed allochrony) is quite common across the tree of life [7], and occurs over different time-scales; from the separation in mating time over a day for different sister species of moth [8] to the several years separat-ing reproduction in broods of periodic cicadas [9]. Indeed, speciation via traits that automatically assort mating (e.g., reproductive timing) are thought to be one of the most likely modes of speciation with gene flow [6], as these traits make no require-ment for linkage between loci under divergent selection from the environment and loci involved in mating, thus permitting divergence despite recombination [10]. Given the commonality of allochrony and the apparent potential of reproductive timing in driving speciation, one might expect speciation driven by timing differences (termed allochronic speciation [7]) to be common. However, examples of allochronic specia-tion are rare in the literature [7], with known examples including the Band-Rumped Storm Petrel [11–14], Japanese Winter Geometrid Moth [15–17], Pine Processionary Moth [18–21], Pink Salmon [22–24], White Mountain Arctic Butterfly [25], Stem Gall-ing Midge [26], Periodic Cicadas [9,27,28], Gall Forming Aphids [29], and Acropora corals [30]. Therefore, understanding the mechanisms that drive divergence in repro-ductive timing at the initial stages of speciation might help to resolve this incongruity.

Several mechanisms have been modeled to explain divergence in reproductive timing with gene flow. Models of similarity-based mating (e.g., [31]) provide a general

mechanism for how divergent selection and frequency dependence on timing might drive speciation, but often overlook species-specific details. More specific to allochronic divergence, Hendry and Day (2005) presented a framework to understand how a species living across a latitudinal cline might become isolated due to adaptations to the local onset of seasons [32]. While this framework addressed the genetic nuances during divergence in reproductive timing, the proposed mechanism did not consider how reproductive timing could lead to allochronic divergence in sympatry. On the other hand, Devaux and Lande (2008) showed that random drift can produce ephemeral speciation in a sympatric population of flowers with heritable reproductive timing [33], though this mechanism did not take into account that reproduction for many organisms is timed to an optimal point in the local environmental cycle rather than drifting neutrally [4,7,34]. More recently, Van Doorn et al. (2025) and Bouinier et al. (2025) both addressed the potential role of male-male competition in driving divergence in reproductive timing and found that it is possible under certain genetic conditions [35,36], particularly when timing was sex-linked [36], highlighting the importance of genomic architecture in determining divergence dynamics. This importance was also demonstrated by Sachdeva and Barton (2017), who showed that in a magic trait model, polygenic traits displayed distinct divergence dynamics as compared to similar models of fewer loci [37].

In contrast to models aimed at explaining the emergence of reproductive divergence, recent work in *C. marinus* has focused on understanding the role of reproductive timing in maintaining stable coexistence between chronotypes in sympatry. Ekrem et al. (2025) explored how mate-finding Allee effects, growth-survival trade-offs, and ontogenetic diet shifts might facilitate coexistence between chronotypes when the environment selects against hybrid phenotypes [38]. They found that strong priority effects made coexistence unlikely, unless there is a suitably timed ontogenetic niche shift [38]. There is, however, currently no empirical data to suggest the occurrence of such an ontogenetic shift [38]. On the other hand, a field study examining two sympatric chronotypes in Roscoff, France, found that the chronotype reproducing during the new moon (Ros-2NM) had a shifted emergence with respect to the new moon spring tide, while emergence for the full moon chronotype (Ros-2FM) was not shifted [39]. As low tide water levels change throughout the lunar month, reaching their minimum during the spring tides, oviposition for Ros-2NM occurs when low tide levels are intermediate, while oviposition for Ros-2FM mates happens when low tides are the lowest [39]. This results in spatial separation of the chronotypes over the bathymetric gradient, providing a mechanism by which differences in reproductive timing can facilitate their coexistence [39].

Here, we build upon this previous work to investigate whether this link between reproductive timing, low tide level, and spatial separation of the larvae is not just a mechanism for coexistence, but a mechanism for allochronic divergence as well. Given the complexity of *C. marinus'* life cycle, we address this via individual-based modeling, replicating *C. marinus'* ecology and reproductive behavior. In our simple representation of the organism, we observe that competition between larvae for space is enough to produce allochronic divergence. We explore how competition strength and dispersal rate affect divergence dynamics, and then expand this model to investigate the potential influence of genomic architecture by incorporating an explicit genetic basis as well as sexual reproduction and recombination. The reproductive timing of *C. marinus* is known to involve several to many genes [40–43], though the genetic basis remains unknown. Therefore, we investigate how changing the number of genes, from monogenic to omnigenic, as well as the distribution of effect sizes for mutations and the heritability of reproductive timing, all influence the potential for divergence. Lastly, we show that our findings are robust to initial conditions and the choice of environmental fitness function. We conclude that divergence is possible over a broad range of biologically feasible conditions, suggesting that this mechanism plausibly drove the divergence of the sympatric chronotypes in Roscoff. This detailed exploration of our model allows us to comment on the potential for competition to drive allochronic divergence more generally, especially for cases in which environmental cycles mediate offspring dispersal.

## Results

### *Clunio marinus* system and model assumptions

*C. marinus* is endemic to rocky stretches of the European Atlantic Coast. Most of their life is spent in the aquatic larval state (between 6 and 14 weeks), feeding on biofilm in the deeper regions of the intertidal zone, while the adult form of *C.*

*marinus* lasts only a few hours and performs no function other than reproduction. The timing of metamorphosis between larval and adult stages is controlled by an endogenous biological clock synchronized to the lunar cycle (called the "circalunar clock"), such that hundreds of thousands of individuals will emerge from the water to mate and oviposit at low tide on spring tide days (Fig 1 bottom right). Mating lasts only a few hours, during which the motile males will seek out freshly emerged wingless and immotile females and take them to the water's edge, where they can oviposit on exposed algae or rocks (Fig 1 top right). *C. marinus* requires these exposed substrates for successful oviposition, as it allows them to glue down their eggs to prevent them from being washed away by the incoming tide. Additionally, in copula, the males are incapable of true flight and instead hover on the water's surface like an airboat. Combining the requirement for an exposed substrate and the inability to fly ensures that ovipositing always occurs on the water's edge at the level of the tides during which the couple is mating (Fig 1 top left). In this way, the timing of reproduction is related to the depth at which the eggs are laid.

We begin by modeling the life cycle of each individual over a single generation (Fig 1, see Methods:Model Overview for a more detailed description). For simplicity, the only spatial component we consider is depth and initially we model reproduction as clonal. Later, we will extend our model to include more realistic reproduction. We assume that the emergence of individuals always happens during the second low tide for a given day, as is the case for all known lunar populations (like Ros-2FM and Ros-2NM) of *C. marinus* [5]. Therefore, under this scenario, the depth at which the offspring live is determined by the depth of the low tide during the parents' lunar day of emergence, plus how far larvae subsequently disperse. We also assume that there is a relationship between the fitness of individuals and the depth at which they lived in the intertidal zone. This is based on the observation that larvae prefer to live in specific species of macroalgae [44]. Macroalgae communities vary dramatically across the bathymetric gradient of the intertidal zone, so the location of *C.*

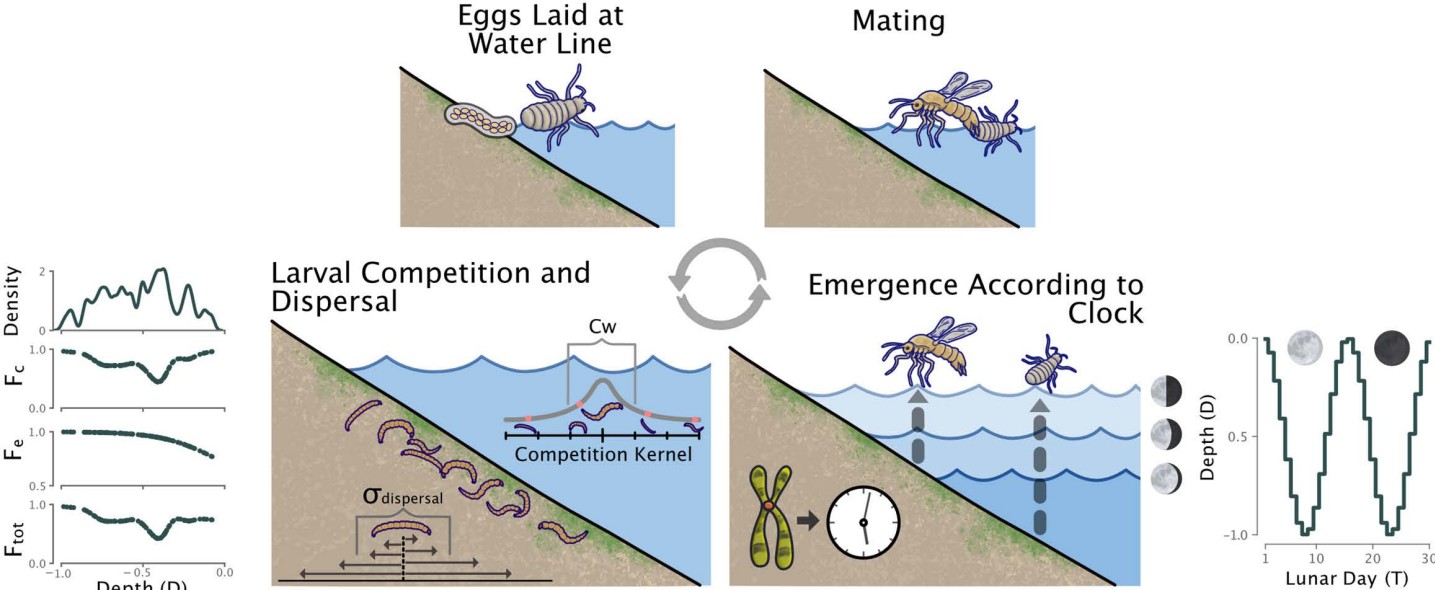

**Fig 1. Schematic of the modeled *C. marinus* behavior.** A simulated generation begins with the adults emerging during low tide on the lunar day determined by their internal circalunar clock. The height of low tide for different lunar emergence days $T$ (i.e., the timing phenotype) is plotted on the lower right. Males and females find each other, and the male takes the female to the waterline for oviposition. The larvae hatch and compete with each other for resources while dispersing throughout the intertidal zone. The dispersal rate is denoted by $\sigma_{dispersal}$ and the competition kernel width $Cw$ determines the strength of competition. The overall fitness of an individual $F_{tot}$ is the product of the environmental fitness component $F_e$, determined by the depth where the individual lives in the intertidal zone, and the competition component $F_c$, which is based on the level of competition an individual experiences as a larva. These fitness functions are plotted on the lower left of the schematic.

*marinus* larvae might also be reflective of the bathymetric ranges for their host macroalgae species. Moreover, since the lower reaches of the intertidal are exposed less by the tides, they might offer a refuge from desiccation. Lastly, we assume that larvae are also competing for space, with competition being the strongest where population density is highest. Larvae living in less dense areas would, therefore, have a fitness benefit over those in more populated areas due to the lack of competition (Fig 1 bottom left).

## Divergence in ecological parameter space

To test if the modeled behavior and ecology could result in allochronic divergence for our simplified model, we simulated a population of *C. marinus* for 500 generations, with an initial phenotype of lunar day 15. The resulting number of chronotypes were attained by counting the number of peaks in the distribution of phenotypes for the population at the end of the run. By plotting the distribution of phenotypes per generation of a single simulation run (Fig 2B), we found that the population quickly branched into two chronotypes, one which emerged during the spring tide and another which emerged just off the neap tide. Along with the differences in emergence timing, when plotting the depths of individuals over generations (Fig 2C), we found that the two chronotypes lived at different depths in the intertidal zone, corresponding to their time of emergence. This result matches observations for the time at which the sympatric chronotypes emerge in the wild and at what depth they are found [39].

To better understand how the ecological parameters of competition kernel width ($Cw$) and dispersal rate ($\sigma_{dispersal}$) affect the dynamics of divergence, we plotted the average number of chronotypes present at generation 500 in 1000 simulation runs for all $Cw/\sigma_{dispersal}$ combinations of 40 values between 0.025 and 1 (Fig 3A). We found that the population diverged into two chronotypes consistently in a large portion of the ecological parameter space and would even diverge into three or more chronotypes for smaller values of $Cw$ and $\sigma_{dispersal}$. This suggests that divergence is a robust outcome

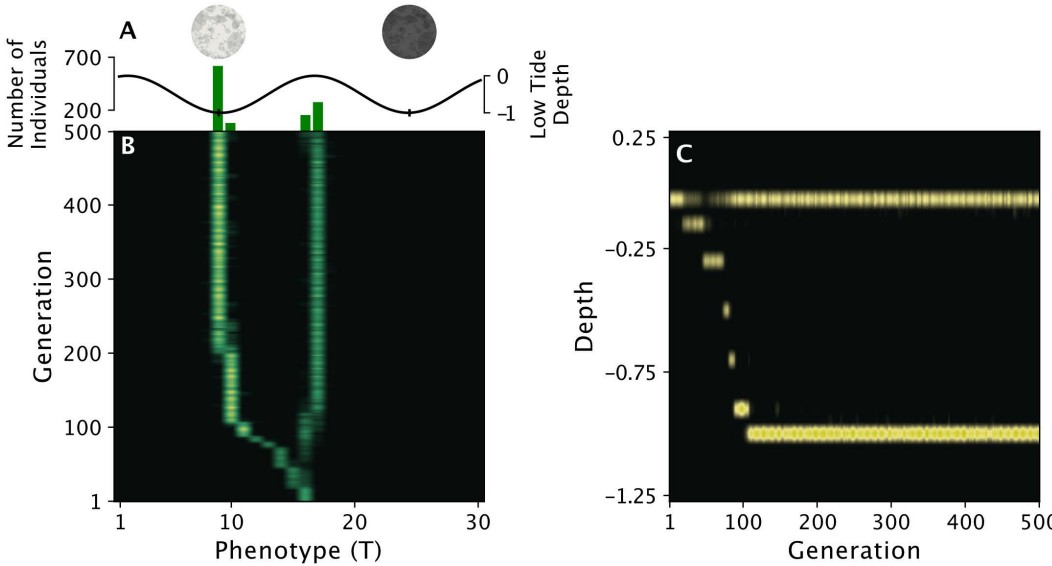

**Fig 2. A single simulation run of a population showing how branching occurs after only a few generations, resulting in one chronotype reproducing during a spring tide and another reproducing just off of the adjacent neap tide.** A: The green bars indicate the number of individuals with each phenotype at generation 500, and the black line denotes the height of low tide for each day in the lunar month. B: Each row is the distribution of phenotypes present at that generation, with brighter green indicating a larger number. Phenotypic divergence was observed after just a few generations. C: This divergence was accompanied by a separation in the depth at which larvae lived in the intertidal zone, with each column now representing one generation. This pattern of separation in depth follows what has been observed for *C. marinus* in the wild. The parameter values used for the simulation are given in Table 1.

**Table 1. Parameters values used for all simulations of the simple version of the model, unless stated in the text.**

| Parameter | Symbol | Value |
|---|---|---|
| Mutation Rate | $\mu$ | 0.18 |
| Competition Kernel Width | $Cw$ | 0.1 |
| Dispersal Rate | $\sigma_{dispersal}$ | 0.25 |
| Carrying Capacity | $K$ | 100 |
| Maximum Number of Offspring | NA | 5 |
| Initial Phenotype | NA | 15 |
| Initial Depth | NA | 0 |
| Initial Population Size | NA | 10 |
| Number of Generations | NA | 500 |

of our model. We also found that if the value of $Cw$ was too large in relation to $\sigma_{dispersal}$, then the population would collapse and result in zero chronotypes at the end of the simulation.

To see how varying these parameters affected the dynamics of divergence, we additionally plotted the change in phenotypes of a single simulation run for three $Cw/\sigma_{dispersal}$ combinations (Fig 3B). For small $Cw/\sigma_{dispersal}$ values, the population would branch rapidly into ephemeral chronotypes. As the values increased, this ephemeral branching was reduced to just a single branching event, and eventually, with large values, no branching occurred. To explore these dynamics more deeply, we plotted the probability of a mutant phenotype taking over a resident population for all phenotype combinations (S1 Fig). This was done by simulating a population with no mutation and introducing a single mutant individual at generation 250. The phenotypes present in the population were then checked at generation 500 to see if the mutant invaded the population, driving the resident phenotype extinct, if the resident phenotype persisted with the mutant unable to invade, or if the mutant and resident phenotype could coexist. This was repeated 1000 times for each phenotype combination. For small $Cw/\sigma_{dispersal}$ values, the majority of combinations could coexist with one another, reflecting the rapid branching seen in the characteristic simulation run. Areas without coexistence were small and could easily be skipped over with a single mutational step. As the values of $Cw/\sigma_{dispersal}$ increased, the areas of coexistence shrank until, eventually, the only mutant strategies able to coexist with a resident were too far to be reached via a single mutational step. In other words, the population would be unable to produce two coexisting chronotypes via mutation, leading to no branching.

Our findings so far show that divergence is possible and even common over the ecological parameter space. This supports the feasibility of the hypothesized mechanism for allochronic divergence in *C. marinus*. However, the applicability of our findings is limited by the simplifying assumptions we made about the mutation and recombination of the trait. The dynamics of speciation are affected by the ecology of an organism and by the genomic basis of the trait under divergent selection [45]. Our simplified mutational process may, therefore, not be biologically realistic. Additionally, while the adaptive dynamics simulations we have employed thus far might predict phenotypic branching, they cannot be used to infer if genetic differentiation following phenotypic branching (i.e., speciation) would occur [46]. These arguments motivated us to extend our model to include an explicit and biologically realistic genetic basis behind the emergence phenotype, sexual reproduction and recombination.

## Adding recombination and a genetic basis

To extend our model, individuals were simulated as sexually reproducing diploids with recombination and a quantitative emergence phenotype produced by an additive genetic basis. This was accomplished by encoding the reproductive ecology of *C. marinus* into SLiM and SLiMgui version 4.0.1 [47], using SLiMs default recombination and mutation

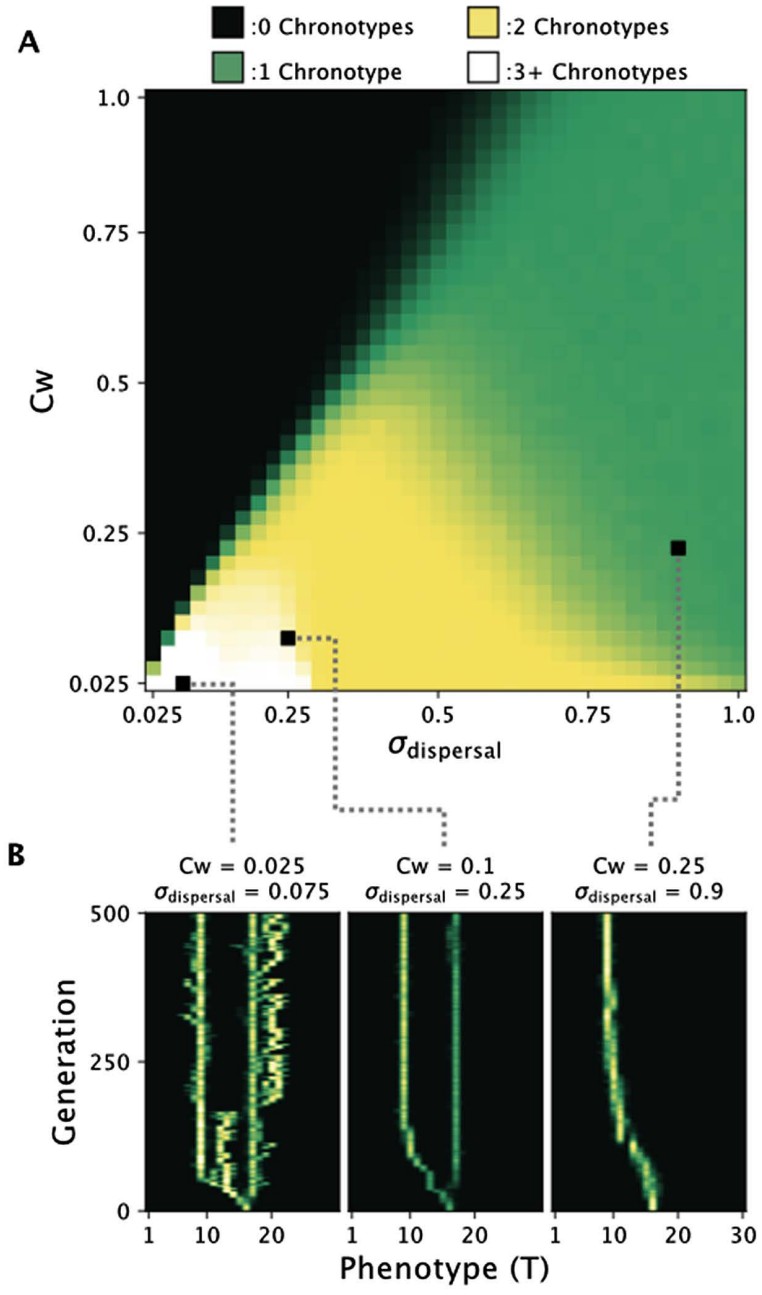

**Fig 3. Exploring how dispersal rate $\sigma_{dispersal}$ and competition strength *Cw* affect divergence in our model of *C. marinus'* reproductive ecology.** A: The average number of chronotypes after 500 generations for 1000 simulations across *Cw*/$\sigma_{dispersal}$ parameter space. Areas where divergence occurs are in yellow and white, and constitute a significant portion of parameter space. Black indicates the region of parameter space in which the population went extinct. B: Single simulation runs under three different parameter combinations of *Cw*/$\sigma_{dispersal}$ showing how changing these parameters affects the dynamics of branching, if it occurs at all.

settings with an equal recombination and mutation rate of $1 \times 10^{-7}$ per base-pair, per generation (see Methods:Extended Model for more details). Unless otherwise stated, values for the parameters in this extended model are listed in Table 2. As with the simpler model, we simulated a population 1000 times for each combination of 40 values of *Cw*

**Table 2. Parameter values used for all simulations of the model with the genetic extension, unless otherwise stated in the text.**

| Parameter | Symbol | Value |
|---|---|---|
| Mutation Rate | $\mu$ | $1 \times 10^{-7}$ |
| Recombination Rate | $r$ | $1 \times 10^{-7}$ |
| Mutation Effect Size Standard Deviation | $\sigma_{effect}$ | 0.5 |
| Proportion of Genome Effecting Trait | $P_{effecting}$ | 0.1 |
| Non-genetic Component of Trait | $\sigma_{emergence}$ | 0.25 |
| Competition Kernel Width | $Cw$ | 0.1 |
| Dispersal Rate | $\sigma_{dispersal}$ | 0.25 |
| Carrying Capacity | $K$ | 200 |
| Maximum Number of Offspring | NA | 10 |
| Initial Phenotype | NA | 15 |
| Initial Depth | NA | 0 |
| Initial Population Size | NA | 10 |
| Number of Generations | NA | 1500 |

and $\sigma_{dispersal}$ between 0.025 and 1. Due to generally slower dynamics in our model with recombination, we now ran each simulation for 1500 generations. The average number of chronotypes at generation 1500 for each $Cw/\sigma_{dispersal}$ combination was then plotted (Fig 4A). Compared to the simpler model's ecological parameter space, the added genetic basis and recombination reduced the area where divergence was possible, both for the area of single branching events (in lilac) and the area of multiple branching events (in white). Meanwhile, the area of extinction remained the same between the models.

To observe the branching dynamics more directly, we again plotted the distribution of phenotypes over time for individual simulation runs for three values of $Cw/\sigma_{dispersal}$ (Fig 4A). For high values, the dynamics resembled those of the simpler model. However, for small values, instead of the rapid and ephemeral branching seen previously, the population split into three stable chronotypes, with the two chronotypes of a singular phenotype and a third intermediate chronotype with a more variable phenotype.

Besides affecting the dynamics of divergence, the inclusion of a genetic basis in the model additionally allowed us to see if genetic divergence could result from the hypothesized mechanism. We recorded the alleles from variable sites (both neutral and non-neutral) present in a random sample of 100 individuals from the population every 200 generations for each of the three simulation runs. From these alleles, we performed principal components analysis and plotted the first two principal components, coloring each individual by the generation in which it was sampled (S2 Fig). We found that phenotypic divergence was reflected in the genetic divergence, with the run at intermediate $Cw/\sigma_{dispersal}$ values producing two genetic clusters corresponding to the two chronotypes, and the run at high $Cw/\sigma_{dispersal}$ values resulting in a single genetic cluster for the single chronotype. The run at low $Cw/\sigma_{dispersal}$ produced three genetic clusters, concordant with the three chronotypes observed, though the distinction between two of the clusters was not clear, indicating that although there was phenotypic divergence between the chronotypes, gene-flow was still ongoing. Overall, these results make clear that the hypothesized mechanism of divergence for *C. marinus* chronotypes can not only drive phenotypic divergence, but can also precipitate genetic divergence as well, meaning that this mechanism could result in speciation. However, these results also emphasize the relevance of the underlying genetic architecture for divergence dynamics. Therefore, we wanted to explore how changing the genetic architecture might affect the feasibility of genetic divergence.

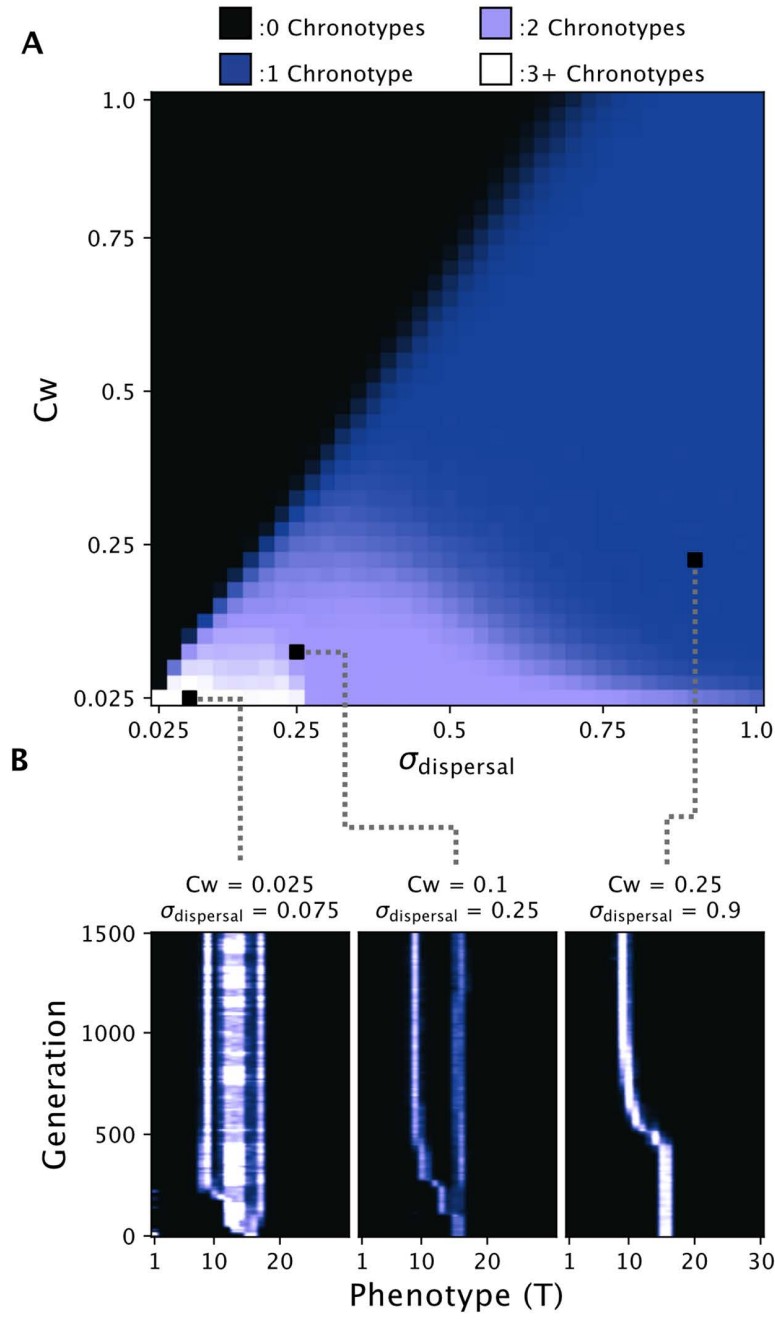

**Fig 4. Divergence across $\sigma_{dispersal}/Cw$ parameter space after extending the model to including realistic sexual reproduction, recombination, and an additive genetic basis A: The average number of chronotypes after 1500 generations for 1000 simulations across $Cw/\sigma_{dispersal}$ parameter space, with areas of divergence in lilac and white.** Despite the extensions, we still find divergence occurring over a range of parameter values. B: Single simulation runs under three different parameter combinations of $Cw/\sigma_{dispersal}$. Branching dynamics are conserved despite the model's extensions, except for low values of $Cw/\sigma_{dispersal}$, where sexual reproduction, recombination, and an additive genetic basis result in more phenotypic stability for chronotypes.

PLOS Computational Biology

## Divergence in genetic parameter space

The genetic architecture underlying the emergence phenotype in our model could be controlled by three parameters: the proportion of the genome that, if mutated, would affect the phenotype ($P_{effecting}$), the standard deviation of effect sizes for those mutations ($\sigma_{effect}$), and the non-genetic component of the trait ($\sigma_{emergence}$) which could be interpreted as the inverse of heritability. We explored the space of these parameters to see where divergence did or did not occur by simulating a population 1000 times for each combination of 21 values between 0–1, 0 to 0.5, and 0–3 for $P_{effecting}$, $\sigma_{effect}$, and $\sigma_{emergence}$, respectively. Since the genetic parameters might impact how quickly the population diverges, we increased the number of generations to 3000 to avoid potentially stopping the simulation prematurely. To test if genetic divergence had occurred in the population at generation 3000, we leveraged the fact that a diverged population will have an excess of alleles segregating at intermediate frequency. We, therefore, recorded the proportion of alleles in the population with a frequency between 0.3 and 0.7 at generation 3000 and plotted the average value over the 1000 runs for each $P_{effecting}$/$\sigma_{effect}$/$\sigma_{emergence}$ combination (Fig 5).

We found that divergence occurred for much of the genetic parameter space when the non-genetic component of the trait, $\sigma_{emergence}$, was less than or equal to 2.25. For higher values of $\sigma_{emergence}$, i.e., at low heritability, no divergence was observed. Within the range of $\sigma_{emergence}$ that could produce divergence, when the effect size of mutations were low ($\sigma_{effect}$ < 0.1), no value of $P_{effecting}$ would result in divergence. In other words, even if the entire genome contributed to the emergence timing trait, if the effect sizes of mutations were too small, divergence would not occur. On the opposite end of the spectrum, when the proportion of loci affecting the trait, $P_{effecting}$, was small, divergence did not occur regardless of the value for $\sigma_{effect}$, or occurred rarely for certain values of $\sigma_{emergence}$. This meant that if the trait was controlled by one or a few genes, then divergence would not occur consistently, even if the effect size of mutations was large.

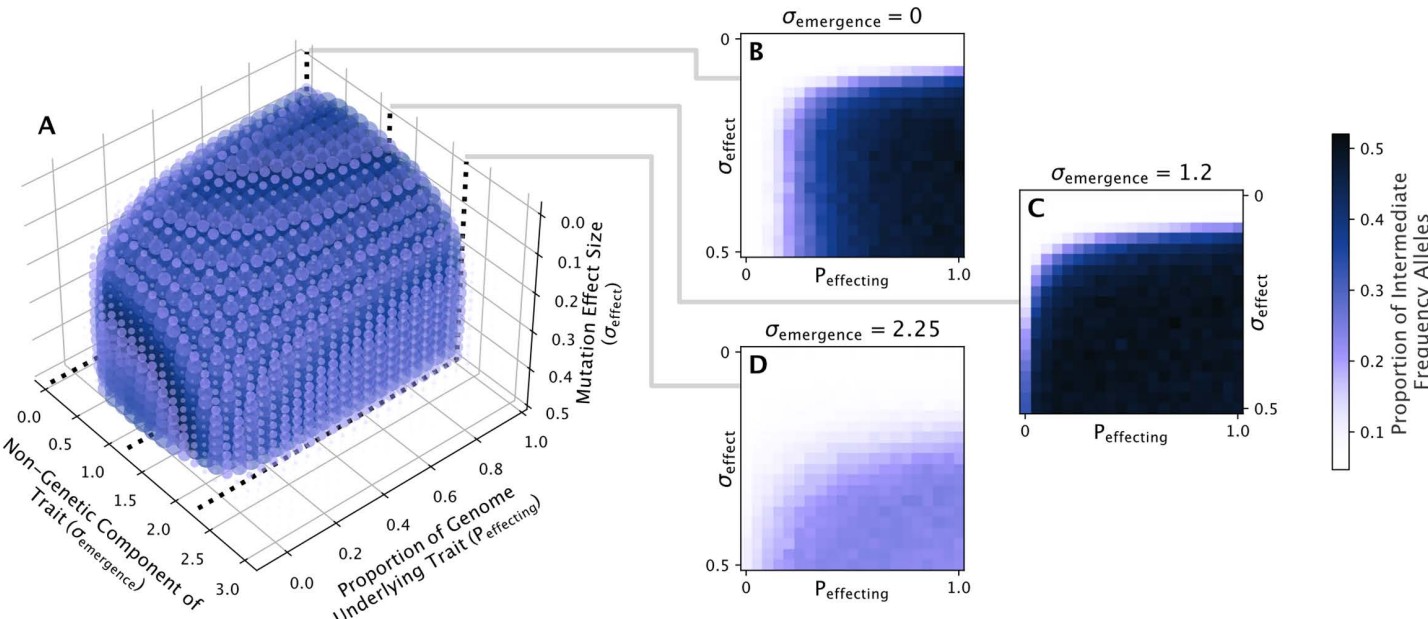

**Fig 5. Divergence across genetic parameter space.** A: A 3D scatter plot of parameter space for the genetic parameters $\sigma_{emergence}$, $P_{effecting}$, and $\sigma_{effect}$. Divergence was observed via an increase in the proportion of intermediate frequency alleles (between 0.3 and 0.7) in the population over the simulation run. Therefore, the color of each dot represents the average over 1000 simulation runs of this value at generation 3000. Each dot has been additionally sized by the average proportion of intermediate frequency alleles so that parameter combinations that did not result in divergence do not obscure ones that did. B,C,D: Slices of the genetic parameter space at three values of $\sigma_{emergence}$ exemplify how $\sigma_{effect}$ and $P_{effecting}$ affect divergence.

Where divergence did occur was above a certain threshold for both $\sigma_{effect}$ and $P_{effecting}$, where there were enough genes and mutations with large enough effect size to result in the population splitting. This threshold was then determined by $\sigma_{emergence}$. For most of the genetic parameter space, increasing $\sigma_{emergence}$ would increase the value of this threshold (Fig 5C vs 5D). However, for values of $\sigma_{emergence}$ approaching zero, this pattern flipped, and decreasing $\sigma_{emergence}$ (i.e., increasing the heritability) resulted in a larger threshold for $P_{effecting}$ required to produce divergence (Fig 5C vs 5B).

In general, these results show that divergence via the hypothesized mechanism does depend on the genetic architecture of the trait, though it occurs readily for most architectures so long as the value of $\sigma_{emergence}$ is low enough. Taken together with the results for the ecological parameter space, it appears that divergence is not a rare outcome, but rather can result over a range of parameter values. What remains to be seen, however, is how the initial phenotype of the population or alternate fitness functions might affect the commonality of divergence.

### Initial phenotypes and alternate fitness functions

To test if divergence occurs regardless of the initial phenotype, and to see where in phenotype space the population branches, we simulated a population 1000 times at each possible initial phenotype (day 1–30) with all other parameters as in Table 1. For each simulation, we recorded the phenotype at branching and the final phenotypes at generation 1500. We then plotted the distribution of both the branching and final phenotypes for each initial phenotype (S3 Fig). We found that regardless of where the population started, branching always occurred. Additionally, though the final phenotypes depended on the initial phenotype, the final phenotypes relative to the tidal cycle were always the same, with one chronotype emerging during a spring tide and the other emerging just off the neap tide.

One last major assumption in our model that may affect whether divergence occurs was that of our chosen environmental fitness function. The sigmoid environmental fitness function that we had used thus far was designed to reflect an increasing favorability towards the bottom of the intertidal zone, though its specification was somewhat arbitrary. The actual relationship between fitness and depth is unknown, and could be influenced by several factors such as food and habitat availability or desiccation risk, which may vary across depth. On the other hand, it could be that all depths are equally habitable, with no relationship between depth and fitness.

To understand how the choice of environmental fitness function affects the dynamics of our model, we again simulated over all combinations of ecological parameters ($Cw$ and $\sigma_{dispersal}$) for four additional fitness functions. These were a flat function representing no relationship between fitness and depth at which larvae lived, a linear function with higher fitness at increasing depth, a unimodal function which represents an optimal depth in the intertidal zone, and a bimodal function for two optimal depths in the intertidal zone (S4 Fig). As before, we simulated a population 1000 times for 1500 generations for each combination of 25 values of $Cw$ and $\sigma_{dispersal}$.

We found that while the fitness function did have some effect on the shape of the regions in ecological parameter space that would result in 1, 2, or 3+chronotypes, none of the fitness functions prevented divergence from occurring (S4 Fig). The most restrictive towards speciation was the linear function, for which divergence would only occur for smaller values of $\sigma_{dispersal}$. Otherwise, changing the fitness function had only a minimal effect on the dynamics of our model. We can, therefore, conclude that our hypothesized mechanism of divergence is robust to the choice of fitness function, even when there is no relationship between depth and fitness. This means that, regardless of what the actual environmental fitness function is for *C. marinus*, it is likely that divergence is possible under a broad set of ecological and genetic parameters.

## Discussion

Allochronic divergence is the process through which one population splits into two due to a shift in reproductive timing, a situation which can arise when this timing is heritable [7]. Our primary question for this study was to determine if allochronic divergence could result from the reproductive ecology of *C. marinus*, producing reproductively isolated chronotypes such as those seen in Brittany, France [5]. Following a previous study that found the depth at which individuals

live in the intertidal zone is related to their reproductive timing [39], we devised an individual-based model to replicate *C. marinus'* reproductive ecology, assuming that the larvae are competing with each other for space. Through replicate simulations of a population under this model, we found that evolutionary branching readily occurred over a range of ecological parameters. We then extended our model to more accurately replicate *C. marinus'* sexual reproduction and recombination, as well as the genetic basis behind their emergence timing phenotype. Through exploring the genetic parameter space, we found that our proposed mechanism still results in divergence despite these extensions, so long as the non-genetic (i.e., non-heritable) component of the trait remains small enough. We additionally showed that the observed divergence was not limited to certain initial phenotypes but rather occurred for all. Lastly, we tested our assumptions about the relationship between fitness and depth and found that our mechanism of divergence was robust to the choice of the environmental fitness function.

As the principal goal of this study was to determine if the modeled behavior could result in divergence, it makes sense to think of what the parameter values might be for *C. marinus* in the wild. For the ecological parameters of $Cw$ and $\sigma_{dispersal}$, in the model with recombination (see Fig 4A) we observed divergence for values of $(Cw, \sigma_{dispersal})$ ranging from $(0.25, 0.25)$ to $(0.025, 1)$. These values, however, are relative to the depth of the intertidal zone. Around Roscoff, this can be on the order of 10 meters in depth, so scaling our parameters to an intertidal zone of 10 meters gives us estimates of (125 cm, 125 cm) to (12.5 cm, 500 cm), with values lower than these resulting in divergence. For larvae that are only a few millimeters long and live a primarily sedentary lifestyle, it is very likely that the actual value for $\sigma_{dispersal}$ would be far lower than these estimates. For the true value of $Cw$, this would depend on the mode of competition for larvae. However, since the larvae spend most of their time in their tubes, it is likely they compete only directly with their neighbors for food and space. Therefore, we expect $Cw$ to be very small as well; smaller than the values which result in divergence for our model.

For the genetic parameters, we can make an estimated guess as to what their values may be based on numerous studies that have probed the molecular basis of *C. marinus'* circalunar clock [40–43]. Several loci of large effect are involved in differences in emergence timing between populations [41], as well as many loci of likely smaller effect that are associated with the loss of the synchronized emergence in populations in the Baltic Sea [43]. Therefore, the extremes of a single locus of large effect and many loci of small effect can be ruled out, putting *C. marinus* in the region of $\sigma_{effect}/P_{effecting}$ space where divergence could occur. For heritability, while an exact value has yet to be estimated for *C. marinus* populations, when we compare the emergence distribution for the sympatric Roscoff chronotypes to our simulated chronotypes, we find that divergence occurs even when the simulated phenotypic variation is far greater than what is seen in the wild [39]. Therefore, we can conclude that this is a biologically feasible mechanism for allochronic divergence.

Given the apparent commonality of divergence over parameter space in our model, one question which arises is why we only find evidence of allochronic divergence in a small region of Brittany in France [5]. A simple explanation could be that other sympatric chronotypes are yet to be found. There is a chance that a chronotype was missed at a sampling location if sampling did not occur during the chronotype's emergence or at the depth where the chronotype's larvae were living. A second explanation could be the aforementioned tidal height. It might be the case that the values of $\sigma_{dispersal}$ and $Cw$ are large enough so that tidal amplitudes lower than those found in Brittany do not provide sufficient space to support sympatric chronotypes. A counterpoint to this, if we assume that previous sampling efforts were able to find all chronotypes at a location, would be that sympatric chronotypes are not found at every site in Brittany, despite the high tidal amplitude [5]. Therefore, the depth of the intertidal zone may not be the only determining factor for the presence of sympatric chronotypes. Another potential explanation is that the ecological opportunity might exist, but populations are lacking in sufficient genetic variation for divergence. Studies on adaptive radiations in Lake Victoria cichlids and Caribbean pupfish both found evidence for introgression at the root of the radiation, concluding that this introgression facilitated subsequent speciation events [48,49]. Though this scenario was not considered in our modeling efforts, it may also be that adaptive introgression in the ancestor of the sympatric populations allowed for divergence to proceed. Further studies looking into the evolutionary history of these chronotypes would be needed to determine this.

 

Though our model was specifically directed at *C. marinus*, our findings are not limited to a single species. More generally, our model represents an organism that times its reproduction, a trait with a heritable and additive genetic basis, and the time at which offspring are born then determines with whom they will be competing. Timing reproduction is common across the tree of life [34], especially for organisms in the ocean, many of which use a circalunar clock like *C. marinus* [50–52]. Indeed, based on the lack of behaviors other than reproduction under the control of these clocks, it may seem as though they evolved almost exclusively for this purpose [50,52]. Our proposed mechanism may, therefore, be of relevance for understanding speciation in these marine organisms, where the homogeneity and fluidity of the environment usually result in high gene flow, a situation which generally precludes speciation via geographic separation [53].

An important complement to our individual-based approach, particularly when considering its applicability to other species, is the possibility of a deterministic framework that permits analytical reasoning about the conditions for allochronic divergence. Under simplifying assumptions such as clonal reproduction, a continuous phenotype space, and a large, well-mixed population, the system we model falls within the scope of adaptive dynamics, using invasion fitness functions to identify evolutionary singular strategies and classify them as either evolutionary attractors or branching points [54]. Such an approach would allow one to derive, rather than simulate, the conditions under which frequency-dependent competition generates disruptive selection sufficient to drive branching in reproductive timing, as has been done for analogous competition-driven divergence in traits such as cell size [55] or studying the evolution of phenomena such as endosymbiosis [56]. However, the extension to sexual reproduction with recombination complicates this analytical program. As shown in this study, The dynamics of multilocus traits under disruptive selection depend sensitively on genetic architecture in ways that resist closed-form treatment. Simulation-based approaches are therefore still the most tractable method for capturing the full biological realism of the *C. marinus* system.

Moving forward, the key consideration when addressing whether competition drove allochronic divergence for a species pair will be determining the ecological link between competition and reproductive timing. For many species, this link may be direct and, therefore, more obvious, like competition for nesting space in the Madeiran Storm Petrel [13]. However, for others, like *C. marinus*, this link may be indirect and far more cryptic, thus requiring a more detailed understanding of the organism's ecology. Further studies may reveal that competition-driven allochronic divergence initiated ecological speciation in other species as well.

## Methods

### Model overview

Individuals were born at the depth where their mother laid the eggs. After birth, individuals dispersed a random amount as larvae, drawn from a normal distribution with standard deviation, $\sigma_{dispersal}$ (Fig 1 bottom left). The environmental component, $F_e$, of each larva's fitness, was then calculated based on the larva's depth (see Methods:Larval Fitness). We start our model exploration with a fitness function reflective of what we assume it to be in the field, with high fitness at the bottom, low fitness at the top, and a smooth transition between the two (i.e., sigmoid). Later, we experiment with our choice of environmental fitness function to test how it effects divergence dynamics.

Next, the interaction strength between each individual and all other individuals in the population was summed (see Methods:Competition), and this sum was then used to calculate the competition component of each larva's fitness, $F_c$. The product between the environmental and competition components was taken to be the total fitness $F_{tot}$ for the individual, ranging from 0 to 1 (see Methods:Larval Fitness, Fig 1 bottom left). Competition between individuals was calculated for all individuals within a generation at once, with non-overlapping generations. Though this is unrealistic, as differences in reproductive timing should lead to staggered cohorts, previous work has found that cohort staggering, and the temporally varying competition it produces, is not enough to facilitate coexistence for *C. marinus* chronotypes [38]. This is because the 6–14 week lifespan of larvae means that there is still significant overlap between different cohorts [38].

Modeling non-overlapping generations, therefore, allowed for a simpler model while precluding any confounding effects from temporally varying competition within a generation.

After competition was calculated, individuals emerged according to their emergence phenotype; an integer $T$ which ranged between 1 and 30. The height of the low tide for $T$ was calculated using equation 1 (see Methods:Environment, Fig 1 bottom right), and individuals were moved to their respective depths to lay their eggs. The number of eggs laid by each individual was determined by multiplying its fitness by 5 (the maximum number of offspring), then rounding down to the nearest integer. While *C. marinus* females can produce 70–150 eggs at a time [57], we limited the maximum number of offspring in our model to 5 for computational tractability.

The offspring then inherited the parents' phenotype, with a chance of mutation. Mutations occurred by drawing a random number from a normal distribution centered at 0.5, adding this number to the parental phenotype, and then rounding the sum down to the nearest integer. This rounding was done as the phenotype (lunar day of emergence $T$) was discrete, which meant that changes to the phenotype could only occur if the random number drawn was greater than 1 or less than 0. The mutation rate could then be set by changing the standard deviation of this normal distribution, with a value of 0.18 selected as it was low enough for the population to reach carrying capacity between mutations, meaning that population dynamic changes could occur between evolutionary changes. As the lunar day is periodic, we used the modulo operator after mutation to ensure that a phenotype of 31 was recoded as 1, and 0 was recoded as 30. Offspring were then born, and the cycle repeated. Since the number of offspring was determined by fitness, the birth rate regulated the population size.

### Extended model

To extend our model to include an explicit genetic basis for lunar emergence, sexual reproduction, and recombination, we leveraged the software SLiM2 to perform forward in time genetic simulations [47]. SLiM2 represents individuals as diploid genomes, where mutations can occur at fixed positions and influence their fitness [47]. Offspring are created through the recombination of parental genomes, where males and females are explicitly modeled, and the number of offspring is influenced by parental fitness. For our simulations, we used SLiM's non-Wright-Fisher model [47] so that the population size was not fixed, but rather regulated by the density of individuals.

As before, the life cycle of individuals started with them being born at the depth where the eggs were laid, followed by larval dispersal. Fitness was then calculated as before (see Methods:Larval Fitness), and individuals emerged according to their emergence phenotype. However, now the emergence phenotype was calculated based on the alleles present in an individual's genome (see Methods:Genomes and Methods:Emergence). Emerged individuals then found mates and reproduced with one another (see Methods:Reproduction), with females laying their eggs at the low tide water line during their emergence. Again, the number of offspring was determined by the fitness of the parents. However, since reproduction now required two individuals, the maximum number of offspring was increased from 5 to 10, maintaining the ratio of 5 offspring per adult. The carrying capacity $K$ was also doubled to match the increase in the number of offspring per generation. Finally, the adults perished, and their eggs hatched, completing one generation.

### Environment

The environment for the individuals consisted of a 1-dimensional slope extending from 1 unit above the mean tide water level to 1 unit below, with the height of low tide ranging from -1–0. Thus, the only spatial coordinate considered was depth. The depth $D$ of the low tide on the lunar day of emergence $T$ (i.e., an individuals phenotype) was given by:

$$D = \frac{1}{2}\left[\sin\left(\frac{2\pi}{15}T - 5\right) - 1\right]$$

(1)

Constants were selected to produce a function resembling the natural undulation of the tides over a 30-day lunar month (Fig 1 bottom right).

## Larval fitness

An individual's fitness was given by two components: the depth at which it lived before emergence, which determined $F_e$, and the strength of competition between individuals, which determined $F_c$ (Fig 1 bottom left). The total fitness $F_{tot}$ was the product of these two components, and was given by:

$$F_{tot} = F_e * F_c = \frac{1}{1 + e^{6D+1}} * \frac{K}{e^{\frac{10I}{K}} + K}$$

(2)

For the environmental component (the left term), the constants in $e^{6D+1}$ were selected so that fitness was maximal at the deepest depth (-1) and minimal at the shallowest depth (1), while remaining relatively low for areas above the neap tide low water level (at 0). While this choice of fitness function is based off the observation that region of the intertidal zone below the neap tide low water line seems more favorable *C. marinus* larvae, the exact choice of constants influencing the fitness function shape was somewhat arbitrary. Later, we changed the environmental component $F_e$ of this fitness function to test how this affects the model's results. These alternate fitness functions were:

$$F_e = 1$$

(3)

$$F_e = -0.25 * D + 0.75$$

(4)

$$F_e = 0.5 * e^{-\frac{(-0.5-D)^2}{(2*0.3)^2}} + 0.5$$

(5)

$$F_e = 0.5 * (e^{-\frac{(-1-D)^2}{(2*0.2)^2}} + e^{-\frac{(-0.15-D)^2}{(2*0.2)^2}}) + 0.5$$

(6)

representing a flat function, a linear function, a unimodal function, and a bimodal function, respectively (S4 Fig)

For the competition component, $I$ was the competition strength felt by a given individual (see Methods: Competition Strength), and a constant of 10 was again chosen so that fitness was close to 1 with minimal competition and near 0 with maximal competition. For computational tractability, we chose a value of 100 for the carrying capacity ($K$) in the simpler ecological model, and 200 in the model with an explicit genetic basis.

## Competition strength

The competition strength $I$ for an individual was calculated via a Gaussian kernel of the form:

$$I = \sum_{i=1}^{n-1} e^{\frac{(D-D_i)^2}{2Cw^2}}$$

(7)

where $D$ was the depth of the focal individual and $D_i$ was the depth of another individual in the population of size $n$. Summing over all other individuals in the population gave the competition strength. $Cw$ was the competition kernel width, which determined how distantly the focal individual could interact with other individuals. Larger values of $Cw$ translated to the focal individual interacting with a greater number of individuals and, therefore, a higher competition strength.

## Genomes

Each individual was represented by a diploid genome of 1 million base pairs in length. Within the genome, mutations were allowed to occur. These mutations could either affect the emergence phenotype or be neutral, meaning they had no effect on the phenotype or the fitness of the individual. The proportion of effecting mutations to neutral mutations was given by $P_{effecting}$. The value of $P_{effecting}$ roughly translated to how much of the genome determined the emergence phenotype (i.e., the number of genes or regulatory elements). Additionally, the mutation rate was uniform across the genome.

The effect size for each mutation was chosen randomly from a normal distribution centered at zero with a standard deviation of $\sigma_{effect}$. Increasing the value of $\sigma_{effect}$ meant that a greater number of mutations would be of a large effect size.

## Emergence

In the simplified model, the emergence phenotype was given by a single value which could mutate and included no environmental variation $\sigma_{emergence}$, i.e., the emergence phenotype was purely genetically determined. For the model with an explicit genetic basis and recombination, the lunar emergence phenotypes for each individual were calculated as the sum of effect sizes for mutations plus a random value drawn from a normal distribution centered at zero with standard deviation of $\sigma_{emergence}$. This random value represented a non-genetic component of the emergence phenotype. Increasing values of $\sigma_{emergence}$ therefore decreased the relative contribution of the genetic component to the emergence phenotype, decreasing the heritability of the trait. Finally, since genomes were initialized with no mutations, a fixed value was added to set the initial phenotype. Since the lunar day is an integer value, the sum was rounded down to the nearest whole number to give the final phenotype $T$.

$$T = \left\lfloor \sum effect\ sizes + random\ value + initial\ phenotype \right\rfloor \tag{8}$$

Importantly, this phenotype was periodic, meaning that a lunar day of 0 was the same as a lunar day of 30, or lunar day 31 was equivalent to lunar day 1. In our model, this periodicity was enforced using the periodic boundaries functionality of SLiM's spatial engine [47].

## Reproduction

Reproduction began by selecting a female from the population as the mother. The sex of each individual was set when the eggs were laid, and the ratio between males and females was 1:1 for each generation. Since adult *C. marinus* live just a few hours, mating can only occur between individuals that emerge on the same day. Therefore, the set of eligible mates was determined by evaluating which males shared the same emergence phenotype as the selected mother. The father was then randomly selected from this set of eligible males. Following the simplified model, the number of offspring was determined by taking the mean fitness between the parents, multiplying by 10, and then rounding down to the nearest whole number. Finally, each offspring's depth was set to the depth of the tide when their parents emerged (rightmost graph, Fig 1). This process was repeated for all females in the population. If a female emerged on a day with no males present, then she would not reproduce. The same was true for males. Recombination was handled by SLiM using the default settings and a recombination rate of $1 * 10^{-7}$.

## Supporting information

**S1 Fig Pairwise invasability and coexistance plots at three $Cw/\sigma_{dispersal}$ values.** White indicates regions where the resident phenotype persists, orange indicates where the mutant phenotype would invade, and green represents regions where the resident and mutant would coexist. The value of each cell is the average of 1000 simulations. (TIFF)

**S2 Fig Principal component analysis of 100 individuals sampled every 100 generations, from the model with a realistic genetic basis, showing how the observed phenotypic divergence is accompanied by genetic divergence (i.e., speciation).** The three plots correspond to the different $Cw/\sigma_{dispersal}$ combinations of the three simulation runs in Fig 4B. At low values, we find three or more genetic clusters at generation 1500 (yellow dots), though these clusters are not necessarily discrete. At intermediate values, we find two discrete clusters, while at high values, there is only one cluster at generation 1500.
(TIFF)

**S3 Fig The branching and final phenotypes of our model.** A: Kernel density estimates of the distribution of branching phenotypes (in green) and the distribution of final phenotypes (in brown). The black line indicates the height at low tide during the lunar month. B: The relationship between initial phenotype and branching/final phenotype. The diagonal (i.e., where initial phenotype = branching/final phenotype) is represented by a white line. Interpolation between cell values in the heat map has been applied to make the pattern more clear.
(TIFF)

**S4 Fig The effect of different environmental fitness ($F_e$) functions on divergence over ecological parameter space.** The fitness functions tested are A: a flat fitness function (i.e., no relationship between fitness and depth), B: a linear fitness function, C: a unimodal fitness function, and D: a bimodal fitness function. The average number of chrono-types for 1000 simulations at generation 1500 for each combination of the competition kernel width $Cw$ and dispersal rate $\sigma_{dispersal}$ is plotted below each fitness function. Divergence was observed (indicated by the lilac and white regions) for every fitness function tested.
(TIFF)

## Author contributions

**Conceptualization:** Alexander G. G. Jacobsen, Tobias S. Kaiser, Chaitanya S. Gokhale.

**Formal analysis:** Alexander G. G. Jacobsen.

**Funding acquisition:** Tobias S. Kaiser.

**Investigation:** Alexander G. G. Jacobsen.

**Supervision:** Tobias S. Kaiser, Chaitanya S. Gokhale.

**Visualization:** Alexander G. G. Jacobsen.

**Writing – original draft:** Alexander G. G. Jacobsen.

**Writing – review & editing:** Alexander G. G. Jacobsen, Tobias S. Kaiser, Chaitanya S. Gokhale.

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
