## [Decision Letter · Decision Letter 0]

29 Jan 2026

PCOMPBIOL-D-25-02324

How Competition can Drive Allochronic Divergence: A Case Study in the Marine Midge, Clunio marinus

PLOS Computational Biology

Dear Dr. Jacobsen,

Thank you for submitting your manuscript to PLOS Computational Biology. After careful consideration, we feel that it has merit but does not fully meet PLOS Computational Biology's publication criteria as it currently stands. Therefore, we invite you to submit a revised version of the manuscript that addresses the points raised during the review process.

We look forward to receiving your revised manuscript.

Kind regards,

Tyler Cassidy

Academic Editor

PLOS Computational Biology

Natalia Komarova

Section Editor

PLOS Computational Biology

**Additional Editor Comments:**

In general, the reviewers appreciated this study and noted only a few areas of potential confusion. In particular, Reviewers 2 and 3 both made relatively minor comments on the clarity of the presentation. Reviewer 3 mentioned the structure of the first section of the results, which captures the biological background and a brief discussion of the model. Here, the authors may want to consider placing the methods before the results, as the model structure/parameterization is important for the results that follow. Similarly, the authors might consider including Tables S1 and S2 (the model parameters) in the Methods, as these parameter values are important aspects of the model. Finally, Reviewer 1 raises important questions on the model structure and justification for some modelling choices in points 2, 4, and 5. I think it is important to further explain these modelling choices before publication.

**Journal Requirements:**

3) We notice that your supplementary Figures, and Tables are included in the manuscript file. Please remove them and upload them with the file type 'Supporting Information'. Please ensure that each Supporting Information file has a legend listed in the manuscript after the references list.

Potential Copyright Issues:

- Figure 1. Please confirm whether you drew the images / clip-art within the figure panels by hand. If you did not draw the images, please provide (a) a link to the source of the images or icons and their license / terms of use; or (b) written permission from the copyright holder to publish the images or icons under our CC BY 4.0 license. Alternatively, you may replace the images with open source alternatives. See these open source resources you may use to replace images / clip-art:

**Reviewers' comments:**

Reviewer's Responses to Questions

**Comments to the Authors:**

Reviewer #1: the peer review is uploaded as an attachment titled Jacobsen_et_al_PLOS-compbio_mating_types_2025_PEERREVIEW.docx.

Reviewer #2: Based on the biology of a marine midge, the authors investigate via modelling the role of competition in allochronic divergence. While their conclusions suggest that divergence should be largely promoted under multiple ecological and genetic conditions, the rarity of chronotypes really found in natural population is honestly discussed.

Although being not expert in modelling, I had a great interest in reading this manuscript and recommended it for publication.

Just, as far I understand the ecology of the studied chronotypes, is “global” sympatry describing fully their spatial distribution, or could it refer to micro-parapatry?

Also, the results are really convincing to favor allochronic divergence. However, the term speciation is sometime used in the text (for instance line 359), without clear explanation about when the authors consider that divergence lead to speciation. In other term, do they consider speciation as an ineluctable final step of the modelled divergence?

Following, few minor comments…

L16 The use of the adverb “However” sounds strange, as the second sentence introduces a concept not simply and directly in opposition or restriction to the first sentence of the abstract.

The paragraphs are easy to follow in this order, but the first paragraph of the “Results” (l 136-167) corresponds to the description of the biological case, which could be in the Introduction. In the same way, the following paragraph (l 168-213) corresponds more to an overview of Material and Methods than to Results.

L 152 the sentence refers to Fig 1 top left (not bottom left).

Reviewer #3: Please note that this manuscript review was prepared in conjunction with a graduate student.

In their manuscript, “How Competition Can Drive Allochronic Divergence: A Case Study in the Marine Midge, Clunio marinus,” Jacobsen et al. describe their computational investigation of allochronic isolation using the marine midge as a model organism. When simulating reproductive ecology of the marine midge in space-limited scenarios, the authors repeatedly found allochronic divergence. This differentiation occurred with a variety of ecological and genetic parameters, suggesting that, with sufficient heritability, allochronic divergence readily occurs in the marine midge. Further, the authors argue that their findings may generalize to many organisms with time-based reproductive strategies.

The authors present a well-supported computational investigation of allochronic isolation in a marine organism, as they carefully describe model assumptions and test many simulations with variable ecological and genetic parameters. As the manuscript uses computational methods to investigate an understudied ecological divergence mechanism, allochronic isolation, and presents a novel, generalizable finding, the study is a strong fit for PLOS Computational Biology. Therefore, we recommend to accept with minor revisions.

Though a strong investigation of allochronic isolation, we note minor opportunities to improve clarity of analyses. First, the authors provide strong justifications for most of their assumptions, but they do not explain the biological relevance of their emergence time assumption (line 154), beyond model simplicity. To strengthen the article, we recommend providing additional context. Second, we recommend stating explicitly the type of simulations being used, in order to make the article more broadly informative to other researchers. Third, to strengthen the literature review of allochronic isolation, we suggest noting the role of allochrony in the ecological divergence of both Rhagoletis fruit flies and their parasitoid wasps. In particular, we note the recent publication by Yee et al. (2025), “Concordance of eclosion life history timing across trophic levels in communities of host plants, fruit flies, and parasitoid wasps in the Pacific Northwest, USA."

**Have the authors made all data and (if applicable) computational code underlying the findings in their manuscript fully available?**

The PLOS Data policy requires authors to make all data and code underlying the findings described in their manuscript fully available without restriction, with rare exception (please refer to the Data Availability Statement in the manuscript PDF file). The data and code should be provided as part of the manuscript or its supporting information, or deposited to a public repository. For example, in addition to summary statistics, the data points behind means, medians and variance measures should be available. If there are restrictions on publicly sharing data or code —e.g. participant privacy or use of data from a third party—those must be specified.requires authors to make all data and code underlying the findings described in their manuscript fully available without restriction, with rare exception (please refer to the Data Availability Statement in the manuscript PDF file). The data and code should be provided as part of the manuscript or its supporting information, or deposited to a public repository. For example, in addition to summary statistics, the data points behind means, medians and variance measures should be available. If there are restrictions on publicly sharing data or code —e.g. participant privacy or use of data from a third party—those must be specified.

Reviewer #1: Yes

Reviewer #2: Yes

Reviewer #3: Yes

PLOS authors have the option to publish the peer review history of their article (what does this mean?). If published, this will include your full peer review and any attached files.). If published, this will include your full peer review and any attached files.

.

Reviewer #1: **Yes:** Xiaoyuan LiuXiaoyuan Liu

Reviewer #2: No

Reviewer #3: No

**Figure resubmission:**
---

## [Decision Letter · Decision Letter 1]

13 Apr 2026

Dear Dr Jacobsen,

We are pleased to inform you that your manuscript 'How Competition can Drive Allochronic Divergence: A Case Study in the Marine Midge, Clunio marinus' has been provisionally accepted for publication in PLOS Computational Biology.

Best regards,

Tyler Cassidy

Academic Editor

PLOS Computational Biology

Natalia Komarova

Section Editor

PLOS Computational Biology

Reviewer's Responses to Questions

**Comments to the Authors:**

Reviewer #1: The revisions the authors have made in light of the reviewer comments has significantly improved the clarity of the manuscript.

**Have the authors made all data and (if applicable) computational code underlying the findings in their manuscript fully available?**

The PLOS Data policy requires authors to make all data and code underlying the findings described in their manuscript fully available without restriction, with rare exception (please refer to the Data Availability Statement in the manuscript PDF file). The data and code should be provided as part of the manuscript or its supporting information, or deposited to a public repository. For example, in addition to summary statistics, the data points behind means, medians and variance measures should be available. If there are restrictions on publicly sharing data or code —e.g. participant privacy or use of data from a third party—those must be specified.requires authors to make all data and code underlying the findings described in their manuscript fully available without restriction, with rare exception (please refer to the Data Availability Statement in the manuscript PDF file). The data and code should be provided as part of the manuscript or its supporting information, or deposited to a public repository. For example, in addition to summary statistics, the data points behind means, medians and variance measures should be available. If there are restrictions on publicly sharing data or code —e.g. participant privacy or use of data from a third party—those must be specified.

Reviewer #1: Yes

PLOS authors have the option to publish the peer review history of their article (what does this mean?). If published, this will include your full peer review and any attached files.). If published, this will include your full peer review and any attached files.

.

Reviewer #1: **Yes:** Xiaoyuan LiuXiaoyuan Liu

---

## [Editor Report · Acceptance letter]

PCOMPBIOL-D-25-02324R1

How Competition can Drive Allochronic Divergence: A Case Study in the Marine Midge, *Clunio marinus*

Dear Dr Jacobsen,

I am pleased to inform you that your manuscript has been formally accepted for publication in PLOS Computational Biology. Your manuscript is now with our production department and you will be notified of the publication date in due course.

With kind regards,

Lilla Horvath
